# SimVLM: Simple Visual Language Model Pretraining with Weak Supervision

**Zirui Wang**[1,2]*,**Jiahui Yu**[2], **Adams Wei Yu**[2], **Zihang Dai**[2], **Yulia Tsvetkov**[3], **Yuan Cao**[2]
[1]Carnegie Mellon University
{ziruiw}@cs.cmu.edu
[2]Google Research, Brain Team
{jiahuiyu,adamsyuwei,zihangd,yuancao}@google.com
[3]University of Washington
{yuliats}@cs.washington.edu

## Abstract

With recent progress in joint modeling of visual and textual representations, Vision-Language Pretraining (VLP) has achieved impressive performance on many multimodal downstream tasks. However, the requirement for expensive annotations including clean image captions and regional labels limits the scalability of existing approaches, and complicates the pretraining procedure with the introduction of multiple dataset-specific objectives. In this work, we relax these constraints and present a minimalist pretraining framework, named **Sim**ple **V**isual **L**anguage **M**odel (**SimVLM**). Unlike prior work, SimVLM reduces the training complexity by exploiting large-scale weak supervision, and is trained end-to-end with a single prefix language modeling objective. Without utilizing extra data or task-specific customization, the resulting model significantly outperforms previous pretraining methods and achieves new state-of-the-art results on a wide range of discriminative and generative vision-language benchmarks, including VQA (+3.74% vqa-score), NLVR2 (+1.17% accuracy), SNLI-VE (+1.37% accuracy) and image captioning tasks (+10.1% average CIDEr score). Furthermore, we demonstrate that SimVLM acquires strong generalization and transfer ability, enabling zero-shot behavior including open-ended visual question answering and cross-modality transfer.

## 1 Introduction

Self-supervised textual representation learning (Devlin et al., 2018; Radford et al., 2018; 2019; Liu et al., 2019; Yang et al., 2019; Raffel et al., 2019; Brown et al., 2020) based on Transformers (Vaswani et al., 2017) has pushed the state of the art on a wide range of natural language processing (NLP) tasks (Rajpurkar et al., 2016; Wang et al., 2018; Sarlin et al., 2020). One successful approach is to first pretrain the model (e.g. BERT) on large-scale unlabled text corpora using masked language modeling (MLM) objective (Devlin et al., 2018), followed by finetuning on downstream tasks. While this pretraining-finetuning paradigm has been widely adopted, recent work on autoregressive language models (LM) (Radford et al., 2019; Brown et al., 2020) such as GPT-3 has shown strong performance without finetuning by utilizing few-shot prompts (Liu et al., 2021), suggesting the text guided zero-shot generalization is a promising alternative.

Motivated by the success of textual representation pretraining, various efforts have been made to build the multi-modal (visual and textual) counterpart. A line of work (Tan & Bansal, 2019; Lu et al., 2019; Li et al., 2019; Chen et al., 2020b; Li et al., 2020; Su et al., 2020; Zhang et al., 2021) has explored vision-language pretraining (VLP) that learns a joint representation of both modalities to be finetuned on vision-language (VL) benchmarks, such as visual question answering (VQA) (Goyal et al., 2017). In order to capture the alignment between images and text, previous methods have extensively exploited two types of human-labeled datasets from multiple sources, which typically consist of the following steps. Firstly, object detection datasets are used to train a supervised

---

*This work was conducted at Google.

object detector (OD) which allows further extracting region-of-interest (ROI) features from images. Next, datasets with aligned image-text pairs are used for MLM pretraining of a fusion model that usually takes as input the concatenation of the extracted ROI features and the paired text. In addition, due to the limited scale of human annotated data, various task-specific auxiliary losses have been introduced in order to improve performance. These design choices complicate the pretraining protocol of VLP, creating a bottleneck for further quality improvement. What is more, such pretraining-finetuning based approaches usually lack the zero-shot capability, just like their language counterparts. In comparison, another line of work (Radford et al., 2021; Ramesh et al., 2021; Jia et al., 2021) utilizes weakly labeled/aligned data crawled from the web to perform pretraining, achieving good performance and certain zero-shot learning capability on image classification and image-text retrieval. Nonetheless, these methods mainly focus on specific tasks of consideration and thus may not serve as a generic pretraining-finetuning representation for VL benchmarks.

In light of these disadvantages of the existing techniques, we are interested in building a VLP model that: (1) can be seamlessly plugged into the pretraining-finetuning paradigm and achieve competitive performance on standard VL benchmarks; (2) does not require a complicated pretraining protocol as in previous methods; and (3) has the potential towards text guided zero-shot generalization in cross-modal settings. To this end, we propose **SimVLM**, standing for **Sim**ple **V**isual **L**anguage **M**odel, which significantly simplifies VLP by *solely* exploiting language modeling objectives on weakly aligned image-text pairs (Jia et al., 2021). In a nutshell, SimVLM consists of the following components:

- **Objective**. It is trained end-to-end from scratch with a single objective of Prefix Language Modeling (PrefixLM), which can not only naturally perform text generation as GPT-3, but also process contextual information in a bidirectional manner as BERT does.
- **Architecture**. The framework employs ViT/CoAtNet (Dosovitskiy et al., 2021; Dai et al., 2021) and directly takes raw images as inputs. These models can also fit the large-scale data and are readily compatible with the PrefixLM objective.
- **Data**. These setups relieve the requirement for object detection and allow the model to utilize the large-scale weakly labeled dataset, which has better potential towards zero-shot generalization.

Not only is SimVLM simpler, requiring neither object detection pretraining nor auxiliary losses, but it also obtains better performance than previous work. Empirically, SimVLM consistently outperforms existing VLP models and achieves new state-of-the-art results on 6 VL benchmarks without additional data nor task-specific customization. Besides, it acquires stronger generalization in visual-language understanding that empowers zero-shot image captioning and open-ended VQA. In particular, SimVLM learns unified multimodal representation that enables zero-shot cross-modality transfer, where the model is finetuned on text-only data and directly evaluated on image-and-text test examples without further training. Our results suggest that generative VLP can not only match existing MLM-based methods on VL tasks but also demonstrate promising zero-shot potential.

## 2 RELATED WORK

Recent years have seen a rapid progress made in vision-language pretraining (Uppal et al., 2020; Han et al., 2021; Khan et al., 2021). While a variety of approaches have been proposed, a large portion of them require object detection for image region feature regression or tagging as part of the pre-training objectives (Tan & Bansal, 2019; Su et al., 2020; Li et al., 2019; Chen et al., 2020b; Gan et al., 2020; Li et al., 2020; Yu et al., 2021; Li et al., 2021; Zhang et al., 2021; Hu et al., 2021; Cho et al., 2021). These methods rely on a strong object detection model like Fast(er) R-CNN (Ren et al., 2015), which is often trained on human annotated data sets like Visual Genome (Krishna et al., 2016). Using such labeled training data as a prerequisite increases the cost of building the training pipeline, and makes the approach less scalable. Some recent efforts have also explored VLP without object detection module (Xu et al., 2021; Kim et al., 2021; Huang et al., 2021), but they only use clean pretraining data with small scales and thus their zero-shot capability is limited.

On the other hand, multiple cross-modality loss functions have been proposed as part of the training objectives, for example image-text matching (Tan & Bansal, 2019; Lu et al., 2019; Xu et al., 2021), masked region classification/feature regression (Tan & Bansal, 2019; Chen et al., 2020b), object

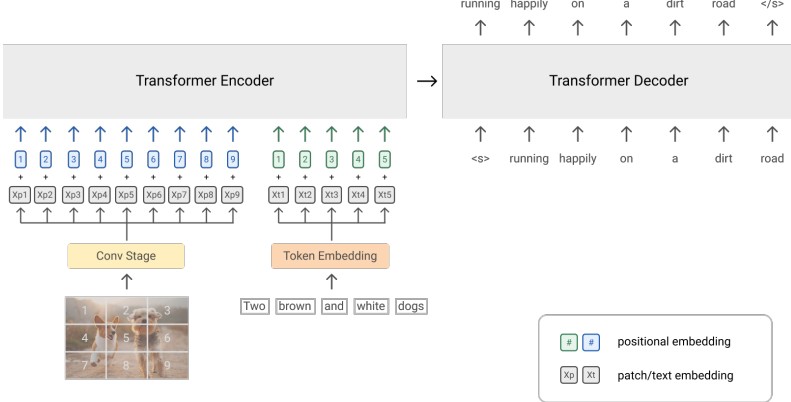

Figure 1: Illustration of the SimVLM model. This shows an example of training with PrefixLM of an image-text pair. For text-only corpora, it is straightforward to remove the image patches and utilize textual tokens only.

attribute prediction (Xu et al., 2021), contrastive loss (Li et al., 2020; 2021), word-region alignment (Chen et al., 2020b) word-patch alignment (Kim et al., 2021). They are often mixed with other objectives including image caption generation and masked language modeling to form compound pre-training losses. This creates the challenge of balancing among different losses and datasets, and thus complicates the optimization procedure.

Our work by contrast, follows a minimalist approach that takes raw image inputs and makes use of only the language modeling loss, without resorting to auxiliary models like faster R-CNN for image region detection. Motivated by recent works (Radford et al., 2021; Ramesh et al., 2021; Jia et al., 2021; Tsimpoukelli et al., 2021) that illustrate zero-shot learning in certain image-text tasks, we train our model using large-scale weakly labeled data only. While concurrent work (Shen et al., 2021) has explored building on top of models pretrained with such dataset, we focus on pretraining from scratch to explore the limit of generative VLP.

## 3 SIMVLM

### 3.1 BACKGROUND

The bidirectional **Masked Language Modeling (MLM)** has been one of the most popular self-supervised training objectives for textual representation learning. As demonstrated by BERT (Devlin et al., 2018), it is based on the idea of denoising autoencoder such that the model is trained to recover the corrupted tokens in a document. Specifically, given a text sequence $\mathbf{x}$, a subset of tokens $\mathbf{x}_m$ are randomly sampled and a corrupted sequence $\mathbf{x}_{\backslash m}$ is constructed by replacing tokens in $\mathbf{x}_m$ with a special [MASK] token. The training objective is to reconstruct $\mathbf{x}_m$ from the context $\mathbf{x}_{\backslash m}$ by minimizing the negative log-likelihood:

$$\mathcal{L}_{\text{MLM}}(\theta) = -\mathbb{E}_{\mathbf{x} \sim D} \left[ \log P_\theta(\mathbf{x}_m | \mathbf{x}_{\backslash m}) \right], \tag{1}$$

where $\theta$ is the trainable parameters of the model and $D$ is the pretraining data. This approach learns contextualized representations that can be further finetuned for downstream tasks. The MLM-style pretraining has been widely adopted in previous VLP models, whereby the input is an image-text pair and the model needs to predict masked tokens by leveraging image ROI features.

Alternatively, the unidirectional **Language Modeling (LM)** trains the model to directly maximize the likelihood of the sequence $\mathbf{x}$ under the forward autoregressive factorization:

$$\mathcal{L}_{\text{LM}}(\theta) = -\mathbb{E}_{\mathbf{x} \sim D} \left[ \log P_\theta(\mathbf{x}) \right] = -\mathbb{E}_{\mathbf{x} \sim D} \left[ \sum_{t=1}^{T} \log P_\theta(\mathbf{x}_t | \mathbf{x}_{<t}) \right]. \tag{2}$$

Compared with MLM, the LM pretraining has also been shown to be highly effective for multiple NLP tasks (Radford et al., 2018). More importantly, it facilitates the model with strong generation

capability that enables text induced zero-shot generalization without finetuning (Brown et al., 2020). While MLM has become the de facto approach in VLP models reviewed above, the generative LM has been understudied.

## 3.2 PROPOSED OBJECTIVE: PREFIX LANGUAGE MODELING

Motivated by the zero-shot capability introduced by pre-training with LM loss, we propose to pretain vision-language representation using the **Prefix Language Modeling (PrefixLM)**. PrefixLM differs from the standard LM such that it enables bi-directional attention on the prefix sequence (e.g. $\mathbf{x}_{<T_p}$ in Eq. (3)), and only conducts autoregressive factorization on the remaining tokens (e.g. $\mathbf{x}_{\geq T_p}$ in Eq. (3)). During pretraining, a prefix sequence of tokens of (a randomly selected) length $T_p$ is truncated from input sequence and the training objective becomes:

$$\mathcal{L}_{\text{PrefixLM}}(\theta) = -\mathbb{E}_{\mathbf{x} \sim D} \left[ \log P_\theta(\mathbf{x}_{\geq T_p} | \mathbf{x}_{<T_p}) \right] = -\mathbb{E}_{\mathbf{x} \sim D} \left[ \sum_{t=T_p}^{T} \log P_\theta(\mathbf{x}_t | \mathbf{x}_{[T_p,t]}, \mathbf{x}_{<T_p}) \right]. \quad (3)$$

Intuitively, images can be considered as prefix for their textual descriptions as they often appear before text in a web document. Therefore, for a given image-text pair, we prepend image feature sequence of length $T_i$ to the text sequence, and enforce the model to sample a prefix of length $T_p \geq T_i$ to calculate LM loss on text data only (an example is shown in Figure 1). Compared to prior MLM style VLP methods, our PrefixLM model under the sequence-to-sequence framework not only enjoys the bidirectional contextualized representation as in MLM, but also can perform text generation similar to LM.

## 3.3 ARCHITECTURE

We adopt Transformer as the backbone of our model due to its success for both language and vision tasks (Devlin et al., 2018; Dosovitskiy et al., 2021). Differently from standard LM, PrefixLM enables bidirectional attention within the prefix sequence, and thus it is applicable for both decoder-only and encoder-decoder sequence-to-sequence language models. In our preliminary experiments, we found that the inductive bias introduced by encoder-decoder model which decouples encoding from generation is conducive to the improvement of downstream task.

An overview of our model architecture is depicted in Figure 1. For the visual modality, inspired by ViT (Dosovitskiy et al., 2021) and CoAtNet (Dai et al., 2021), our model receives the raw image $\mathbf{x} \in \mathbb{R}^{H \times W \times C}$ and maps it into flattened 1D sequence of patches $\mathbf{x}_p \in \mathbb{R}^{T_i \times D}$ as input for the transformer, where $D$ is the fixed hidden size of the transformer layers and $T_i = \frac{HW}{P^2}$ is the length of the image tokens for a given patch size $P$. Following Dai et al. (2021), we use a convolution (Conv) stage consist of the first three blocks of ResNet (He et al., 2016) to extract contextualized patches, which we find advantageous over the naive linear projection (equivalent to $1 \times 1$ Conv layer) used in ViT, consistent with the observation from (Xiao et al., 2021). For the textual modality, we follow the standard practice to tokenize the input sentence into sub-word tokens (Kudo & Richardson, 2018), and the embeddings are learned for a fixed vocabulary. To retain positional information, we add two trainable 1D positional embeddings for image and text inputs separately, and we additionally add 2D relative attention for the image patches within transformer layers (Dai et al., 2021). Notice that we do not add extra modality type embeddings for which we found no improvement in our experiment. We study the effects of various components of the model in Section 4.4.

## 3.4 DATASETS

Since our approach does not rely on an object detection module and only operates with raw image patch inputs, we pretrain all model parameters from scratch using large-scale noisy image-text data, which has better potential for zero-shot generalization. Specifically, we use the image and alt-text pairs introduced in Jia et al. (2021), which are crawled from the web with minimal post-processing. On the other hand, our formulation of PrefixLM is modality-agnostic and thus we can additionally include text-only corpora to compensate for noisy text supervision in the alt-text data. As shown later in our experiments, this unified PrefixLM formulation reduces the modality discrepancy and improves the model quality.

| | VQA | | NLVR2 | | SNLI-VE | | CoCo Caption | | | | NoCaps | | Multi30k |
|---|---|---|---|---|---|---|---|---|---|---|---|---|---|
| | test-dev | test-std | dev | test-P | dev | test | B@4 | M | C | S | C | S | En-De |
| *Base-sized Models* | | | | | | | | | | | | | |
| LXMERT | 72.42 | 72.54 | 74.90 | 74.50 | - | - | - | - | - | - | - | - | - |
| VL-T5 | - | 70.30 | 74.6 | 73.6 | - | - | - | - | 116.5 | - | - | - | 45.5 |
| SOHO | 73.25 | 73.47 | 76.37 | 77.32 | 85.00 | 84.95 | - | - | - | - | - | - | - |
| SimVLM$_{base}$ | 77.87 | 78.14 | 81.72 | 81.77 | 84.20 | 84.15 | 39.0 | 32.9 | 134.8 | 24.0 | 94.8 | 13.1 | 46.6 |
| *Large-sized Models* | | | | | | | | | | | | | |
| UNITER | 73.82 | 74.02 | 79.12 | 79.98 | 79.39 | 79.38 | - | - | - | - | - | - | - |
| OSCAR | 73.61 | 73.82 | 79.12 | 80.37 | - | - | **41.7** | 30.6 | 140.0 | 24.5 | 80.9 | 11.3 | - |
| Villa | 74.69 | 74.87 | 79.76 | 81.47 | 80.18 | 80.02 | - | - | - | - | - | - | - |
| UNIMO | 75.06 | 75.27 | - | - | 81.11 | 80.63 | 39.6 | - | 127.7 | - | - | - | - |
| VinVL | 76.56 | 76.60 | 82.67 | 83.98 | - | - | 41.0 | 31.1 | 140.9 | 25.2 | 92.5 | 13.1 | - |
| SimVLM$_{large}$ | 79.32 | 79.56 | 84.13 | 84.84 | 85.68 | 85.62 | 40.3 | 33.4 | 142.6 | 24.7 | 108.5 | 14.2 | 47.5 |
| *Huge-sized Models* | | | | | | | | | | | | | |
| SimVLM$_{huge}$ | **80.03** | **80.34** | **84.53** | **85.15** | **86.21** | **86.32** | 40.6 | **33.7** | 143.3 | 25.4 | **110.3** | **14.5** | **47.6** |

Table 1: Single model results for vision-language pretraining methods on popular VL banchmarks. We report vqa-score for VQA, accuracy for NLVR2 and SNLI-VE, BLEU@4 for Multi30k and various metrics for image captioning (B@4: BLEU@4, M: METEOR, C: CIDEr, S: SPICE).

Compared to prior VLP methods consisting of two pretraining stages and multiple auxiliary objectives, our model only requires one-pass pretraining using a single language modeling loss in an end-to-end manner, hence the name Simple Visual Language Model (SimVLM).

## 4 EXPERIMENTS

We conduct systematic experiments on a diversified set of visual-linguistic benchmarks, including visual question answering, image captioning, visual reasoning, visual entailment, and multimodal translation. We not only examine our model as a general-purpose VL representation learning in the pretraining-finetuning paradigm, but also study its zero-shot generalization towards open-ended VL understanding.

### 4.1 SETUP

Our models are implemented with the Lingvo framework (Shen et al., 2019). We follow the setup in ViT (Dosovitskiy et al., 2021) to explore 3 variants of SimVLM, namely "Base", "Large", and "Huge", such that each variant follows the same setting as its corresponding ViT variant. All models are pretrained from scratch for about 1M steps on the training set of ALIGN (Jia et al., 2021) and the Colossal Clean Crawled Corpus (C4) dataset presented in Raffel et al. (2019). We mix the two pretraining datasets within each batch, which contains 4,096 image-text pairs (ALIGN) and 512 text-only documents (C4), sharded across 512 TPU v3 chips (Jouppi et al., 2017). More pretraining settings are detailed in Appendix B.1.

After pretrained, our model is finetuned and evaluated on six vision-language benchmarks, including three discriminative tasks: VQA v2 (Goyal et al., 2017), SNLI-VE (Xie et al., 2019), and NLVR2 (Suhr et al., 2018); as well as three generative tasks: CoCo captioning (Chen et al., 2015), NoCaps (Agrawal et al., 2019), and Multi30k (Elliott et al., 2016). We additionally examine its zero-shot generalization and performance on single-modality tasks. Details of tasks considered and the fine-tuning process are outlined in Appendix B.2.

### 4.2 COMPARISON WITH EXISTING APPROACHES

To examine the quality of vision-language pretraining, we first compare SimVLM on the popular multi-modal tasks with state-of-the-art (SOTA) VLP methods including LXMERT (Tan & Bansal, 2019), VL-T5 (Cho et al., 2021), UNITER (Chen et al., 2020b), OSCAR (Li et al., 2020), Villa (Gan et al., 2020), SOHO (Huang et al., 2021), UNIMO (Li et al., 2021), and VinVL (Zhang et al., 2021).

As can be seen in Table 1, SimVLM outperforms all existing models and achieves new SOTA results on all tasks considered, often by a significant margin. This demonstrates our generative pretraining approach is competitive with MLM-based models and that simple framework with weak supervision is sufficient to learn high-quality multi-modal representations.

| | Setup | CoCo Caption | | | | NoCaps | | | |
|---|---|---|---|---|---|---|---|---|---|
| | | B@4 | M | C | S | In | Near | Out | Overall |
| BUTD[a][†] | supervised | 36.3 | 27.7 | 120.1 | 21.4 | - | - | - | - |
| AoANet[b][†] | | 39.5 | 29.3 | 129.3 | 23.2 | - | - | - | - |
| M2 Transformer[c][†] | | 39.1 | 29.2 | 131.2 | 22.6 | 81.2 | - | 69.4 | 75.0 |
| SimVLM$_{base}$ | zero-shot | 9.5 | 11.5 | 24.0 | 7.5 | 83.2 | 84.1 | 82.5 | 83.5 |
| SimVLM$_{large}$ | | 10.5 | 12.0 | 24.9 | 8.3 | 97.6 | 96.5 | 96.3 | 96.6 |
| SimVLM$_{huge}$ | | 11.2 | 14.7 | 32.2 | 8.5 | 101.2 | 100.4 | 102.3 | 101.4 |
| SimVLM$_{base}$ | few-shot | 34.7 | 29.2 | 118.7 | 21.9 | 95.0 | 91.9 | 98.5 | 93.7 |
| SimVLM$_{large}$ | | 35.4 | 30.2 | 124.1 | 22.7 | 102.5 | 100.9 | 106.0 | 102.2 |
| SimVLM$_{huge}$ | | 36.8 | 31.5 | 131.3 | 24.0 | 111.8 | 110.6 | 111.0 | 110.4 |
| OSCAR[†] | pretrain-finetune | **41.7** | 30.6 | 140.0 | 24.5 | 85.4 | 84.0 | 80.3 | 83.4 |
| VinVL[†] | | 41.0 | 31.1 | 140.9 | 25.2 | 103.7 | 95.6 | 83.8 | 94.3 |
| SimVLM$_{huge}$ | | 40.6 | **33.7** | **143.3** | **25.4** | **113.7** | **110.9** | **115.2** | **112.2** |

Table 2: Image captioning results on CoCo Karpathy-test split and NoCaps validation split. For No-Caps, {In, Near, Out} refer to in-domain, near-domain and out-of-domain respectively. [†] indicates Cider optimization. Model references: [a]Anderson et al. (2018) [b]Huang et al. (2019) [c]Cornia et al. (2020).

For the discriminative tasks, the SimVLM$_{base}$ already outperforms all prior methods while using less capacity, and the SimVLM$_{huge}$ obtains almost 4 points absolute score improvement compared to the previous SOTA (VinVL), pushing the single model performance above 80% on VQA for the first time. In addition, SimVLM also consistently outperforms prior methods on NLVR2 and SNLI-VE, illustrating its capability of processing more complex visual-linguistic reasoning. For the generation tasks including image captioning and image translation, SimVLM also shows large improvements using naive finetuning techniques. Our model outperforms on 3 out of 4 metrics on the public "Karpathy" 5k test split of CoCo captioning as well as the NoCaps benchmark than prior methods trained with more complex reinforcement learning approach of CIDEr optimization (Rennie et al., 2017). Finally, SimVLM is also effective for image translation of Multi30k from English to German. These experiments demonstrate that our model can be seamlessly plugged into the pretraining-finetuning paradigm with superior performance, utilizing minimalist pretraining and finetuning procedures.

## 4.3 ZERO-SHOT GENERALIZATION

A crucial benefit of generative modeling and scaling with weak supervision is the potential of zero-shot generalization. Models (Brown et al., 2020; Radford et al., 2021; Jia et al., 2021) have been shown capable of performing few-shot or zero-shot transfer from pretrained models to downstream datasets, even across language boundaries (Lample & Conneau, 2019). In this section, we show-case three different settings of zero-shot applications less explored in prior VLP work, including transferring to unseen tasks, modalities and/or testing instances.

### 4.3.1 ZERO-SHOT/FEW-SHOT IMAGE CAPTIONING

The pretraining procedure of SimVLM can be interpreted as a noisy image captioning objective on real-world web corpus. Thus, it is natural to ask how well this caption ability generalizes to other datasets in a zero-shot/few-shot manner. To this end, we take the pretrained SimVLM model, and directly decode on image captioning benchmarks for the zero-shot setting while finetune on 1% training data for 5 epochs for the few-shot setting. We also found that using a prefix prompt "A picture of" improves the quality of decoded captions, similar to the finding in Radford et al. (2021).

As shown in Table 2, the zero-shot/few-shot performance (Appendix D) of SimVLM is competitive with fully supervised baselines on CoCo, and it also demonstrates strong generalization on the concept-rich NoCaps benchmark by achieving better scores than pretrained models. Figure 2 (a) illustrates sample captions generated by our model (Appendix A). SimVLM is able to not only capture real-world concepts but also provide a detailed description of the visual input. For example, the decoded samples are able to explain complex scenes with multiple objects (e.g. "people", "table with drinks", "dark restaurant"). Besides, the model also shows understanding of fine-grained abstraction such as specific car brand and model (e.g. "Aston Martin", "Vantage"). SimVLM even performs

| | SNLI-VE (T) | SNLI-VE SNLI $\text{Acc}_{dev}/\text{Acc}_{test}$ | MNLI | Multi30k Multi30k (T) | |
|---|---|---|---|---|---|
| | | | | B@4 | M |
| Fully Supervised Baseline | | | | | |
| EVE-Image | | 71.56 / 71.16 | | - | - |
| UNITER | | 78.59 / 78.28 | | - | - |
| SOHO | | 85.00 / 84.95 | | - | - |
| LIUM[a] | | - | | 23.8 | 35.1 |
| GroundedTrans[a] | | - | | 15.8 | 31.2 |
| Zero-Shot Cross-Modality Transfer | | | | | |
| SimVLM$_{base}$ | 71.35 / 71.02 | 72.65 / 72.24 | 64.37 / 63.98 | 15.0 | 24.8 |
| SimVLM$_{large}$ | 72.85 / 72.44 | 73.62 / 73.23 | 66.97 / 66.31 | 17.7 | 30.1 |
| SimVLM$_{huge}$ | 73.56 / 73.08 | 74.24 / 73.86 | 67.45 / 66.97 | 18.2 | 32.6 |

Table 3: Zero-shot cross-modality transfer results on SNLI-VE and Multi30k. For SNLI-VE, the zero-shot model is finetuned on three source datasets: text-only SNLI-VE (Xie et al., 2019), SNLI (Bowman et al., 2015), and MNLI (Williams et al., 2017). For Multi30k, the model is finetuned on text-only Multi30k data. Model reference: [a](Specia et al., 2016).

robustly on challenging images that could be tricky for human, such as abstract or dark pictures. These all illustrate that our model learns a wide range of real-world concepts that generalize well in a zero-shot manner.

### 4.3.2 Zero-shot cross-modality Transfer

Existing pretraining methods have been shown to be successful in transferring knowledge across heterogeneous data spaces. For example, multilingual language models (Devlin et al., 2018; Lample & Conneau, 2019) enable zero-shot cross-lingual transfer such that the model is only finetuned using training data from a source language (typically English) and evaluated on the target language without further training. Inspired by this setup, we explore a novel zero-shot cross-modality transfer paradigm of utilizing VLP models, and evaluate how well our model generalizes across modalities. Since text training data are usually cheaper to obtain compared to visual data, we finetune SimVLM on text-only downstream data and then directly evaluate the zero-shot transfer on joint VL tasks.

Specifically, We utilize SNLI-VE and Multi30k to examine the zero-shot transfer performance. For SNLI-VE, we finetune on three text-only NLI datasets such that the premise sentence is used as the encoder's input while the hypothesis is fed to the decoder, and a similar classifier head is trained on the embedding of the last token in the decoder. At inference, the finetuned model is evaluated by taking the premise image as the encoder input and the corresponding hypothesis sentence to the decoder. As shown in Table 3, SimVLM performs competitively with fully supervised baselines including UNITER under the zero-shot setting. As a sanity check, we also mask out the image feature to predict using the hypothesis only, and find our models can only obtain results close to random guess (average scores of 34.31 / 34.62). This results in performance close to random guess hence demonstrating the effectiveness of SimVLM's cross-modality transfer ability.

In addition, SimVLM is also capable of domain adaption by transferring from the MNLI dataset to SNLI-VE, whereby data comes not only from a different modality but also another domain. We also find it possible to transfer across different languages and modalities using SimVLM. Specifically, we utilize the German image captioning task from WMT 2016 of Multi30k for evaluation, where our model is finetuned on English-German text-only translation data followed by decoding with image-only input in the encoder. Table 3 shows that SimVLM is capable of transferring knowledge across modalities and languages in generative tasks, achieving comparable performance to supervised baselines (decoded examples shown in Figure 2 (b)). These results suggest zero-shot cross-modality transfer emerges with the scaling of weakly labeled data.

### 4.3.3 Open-ended VQA

On the VQA benchmark, the best performing models to date formulate the problem as a discriminative task of multi-label classification over a predefined 3,129 answer candidates, often consisting of short factual terms. In real-world applications, however, it is hard to define a closed set of candidate answers that covering all possible scenarios, making the true open-ended VQA a challenging setup.

| | Dev | Karpathy-test | | | Partial Train | | |
|---|---|---|---|---|---|---|---|
| | | In-domain | Out-domain | Overall | In-domain | Out-domain | Overall |
| Discriminative | | | | | | | |
| UNITER | - | 74.4 | 10.0 | 70.5 | - | - | - |
| VL-T5 | - | 70.2 | 7.1 | 66.4 | - | - | - |
| VL-BART | - | 69.4 | 7.0 | 65.7 | - | - | - |
| SimVLM$_{base}$ | 73.8 | 79.0 | 16.7 | 75.3 | 78.4 | 10.3 | 70.5 |
| SimVLM$_{large}$ | 76.0 | 80.4 | 17.3 | 76.7 | 79.5 | 11.0 | 71.8 |
| SimVLM$_{huge}$ | **76.5** | **81.0** | 17.5 | **77.2** | **80.2** | 11.1 | 72.2 |
| Generative | | | | | | | |
| VL-T5 | - | 71.4 | 13.1 | 67.9 | - | - | - |
| VL-BART | - | 72.1 | 13.2 | 68.6 | - | - | - |
| SimVLM$_{base}$ | 73.2 | 78.3 | 25.8 | 75.2 | 77.1 | 27.1 | 71.3 |
| SimVLM$_{large}$ | 75.2 | 79.5 | 29.6 | 76.5 | 78.7 | 28.4 | 72.5 |
| SimVLM$_{huge}$ | 75.5 | 79.9 | **30.3** | 77.0 | 79.1 | **28.8** | **73.0** |

Table 4: Comparison of discriminative and generative VQA methods. "Dev" refers to standard vqa-score on the VQA validation split. "Karpathy-test" is the setup used in Cho et al. (2021) for evaluation on the Karpathy split with rare answers. "Partial Train" refers to train the model only on partial training data which contain subset of all candidate answers.

Generative models such as SimVLM provide an alternative solution towards this challenge by generating free-form textual answers without being constrained to predefined answers. To this end, we finetune SimVLM using the PrefixLM loss described above where we treat the concatenation of the image and the question as the prefix, and train the model to generate answers.

We then compare the generative approach with classification methods in Table 4. Firstly, we follow Cho et al. (2021) and evaluate model performance on questions with rare answers in the Karpathy-test split. Here, out-of-domain questions are defined as those with best-scoring answer not included in the 3,129 candidates. Results show that SimVLM outperforms both discriminative and generative baselines on all splits. More importantly, the generative SimVLM significantly improves on the out-of-domain split by over 17 points, demonstrating its strong generalization. However, this setup mainly focuses on rare answers and it remains unclear how well the model generalizes to common unseen answers. We therefore proceed to investigate a more challenging setup where we randomly select 2,085 (about two-thirds of 3,129) in-domain answers and partition both train and validation sets into two splits based on whether their best-scoring answers are included in the selected set or not. We then only finetune SimVLM on the in-domain split of the train set and evaluate on the entire validation set. The "Partial Train" column in Table 4 shows that the generative SimVLM is also competent in this setup by scoring reasonably well on over 1,000 unseen answers. Overall, we found the generative SimVLM performs competitively with its discriminative counterpart in the standard setup, and works generally better in the out-of-domain case.

| Method | Acc@1 |
|---|---|
| SimCLRv2 (Chen et al., 2020a) | 79.8 |
| DINO (Caron et al., 2021) | 80.1 |
| CLIP (Radford et al., 2021) | 85.4 |
| ALIGN (Jia et al., 2021) | **85.5** |
| SimVLM$_{base}$ | 80.6 |
| SimVLM$_{large}$ | 82.3 |
| SimVLM$_{huge}$ | 83.6 |

Table 5: Linear evaluation on ImageNet classification, compared to state-of-the-art representation learning methods.

Note that we use the exact matching between generated answers and human labels for score calculation in the above experiment, however it is possible that the model generates appropriate answers in different formats or synonyms. Therefore, in addition to the quantitative study above, we show qualitative generation results in Figure 2 (c). It can be observed that SimVLM is able to generate answers not included in the 3,129 candidate set (e.g. "surgeon" and "wood carving"), demonstrating that SimVLM can transfer knowledge from the pretraining corpus to VQA. It is thus natural to ask whether SimVLM can perform zero-shot VQA without finetuning at all. In our experiments, we found that SimVLM is able to "answer" by completing prompting sentences, as shown in Figure 2 (d). Nonetheless, we also observed that the model falls short in generating meaningful answers to the real questions. We hypothesize that this is due to the low quality of the pretraining data in which most textual descriptions are short and noisy. To verify our assumption, we continue the pretraining process on the cleaner WIT dataset (Srinivasan et al., 2021) for 50k steps. Examples in Figure 2 (e)

show that open-ended VQA ability emerges in SimVLM such that it can generate related responses after finetuning on the knowledge-rich wikipedia dataset.

## 4.4 ANALYSIS

**Single-Modality Tasks.** Since SimVLM performs well on joint vision-language benchmarks, it is natural to ask how well the learned representations perform on tasks of single modality. We hope to gain deeper insights into the model behavior by examining its performance on these benchmarks, but it is not our intention to achieve state-of-the-art on single-modality tasks. In Table 7 (Appendix C), we compare SimVLM with existing VLP models on the GLUE benchmark (Wang et al., 2018), where we mainly follow the text processing procedure in Raffel et al. (2019) and train our model to classify the fully formatted input without token type embeddings. SimVLM performs better than existing VLP methods and competitively with BERT, indicating that it has good language understanding ability. Additionally, we also compute the top-1 accuracy on ImageNet following the linear evaluation protocol in Table 5. Note that our model is not pretrained with a discriminative task such as the contrastive loss, hence we use an average pooling of encoder outputs as image features. Results verify that our model has also learned high-quality image representation.

| Method | VQA score |
|---|---|
| No Pretraining | 49.70 |
| Decoder-only | 65.23 |
| w/ LM | 64.48 |
| SimVLM$_{small}$ | 67.43 |
| w/o Image2Text | 49.23 |
| w/o Text2Text | 65.25 |
| w/o conv stage | 63.11 |
| w/ span corruption | 66.23 |
| w/ 2 conv blks | 65.57 |
| w/ 4 conv blks | 66.55 |
| w/ 10% ALIGN | 66.71 |
| w/ CC-3M | 63.32 |

Table 6: Ablation study on VQA. "w/ LM" and "w/ span corruption" denote replacing the proposed PrefixLM loss with a different pretraining objective. "Image2Text" and "Text2Text" refer to the noisy image-text data and the text-only data used for pretraining. "conv blks" denotes number of ResNet blocks.

**Ablation Study.** To study the contributions from each model component, we conduct ablation study on SimVLM$_{small}$ models with an embedding dimension of 512 and 8 layers. We make comparisons on VQA in Table 6. First, we compare encoder-decoder models with decoder-only models of comparable model size, and find that decoder-only model performs significantly worse on VQA. This suggests the inductive bias of separating bidirectional encoding from unidirectional decoding is beneficial for joint VL representation learning. Next, we study the effectiveness of pretraining objectives and results show that the PrefixLM objective outperforms both span corruption (Raffel et al., 2019) and naive LM, illustrating the importance of using a unified objective formulation for both image-text and text-only data. Moreover, we ablate the contribution of datasets. While weakly aligned image-text data are required for bridging the gap between visual and textual representations, text-only corpora also improves the model quality. This is probably because textual signals are extremely noisy in the former and thus the model relies on the later to acquire better language understanding. In addition, we experimented with 10% ALIGN and CC-3M (Sharma et al., 2018) datasets, and confirms the importance of data scaling. We then study the effect of the convolution stage and find it critical for VL performance. Following Dai et al. (2021), we experiment with using either the first 2/3/4 ResNet Conv blocks, and empirically observe that the 3 conv block setup works best. This indicates that image and text have different levels of representation granularity and thus utilizing contextualized patches is beneficial.

## 5 CONCLUSION

In this work, we present a simple yet effective framework of vision-language pretraining. Unlike prior works using object proposal systems and auxiliary losses, our model processes whole image as patches and is trained end-to-end with a single prefix language modeling objective. Our work suggests a promising alternative to existing VLP paradigm and we hope our work may inspire future research on generative VLP.

ACKNOWLEDGMENTS

We would like to thank Hieu Pham, Chao Jia, Andrew Dai, Bowen Zhang, Zhifeng Chen, Ruoming Pang, Douglas Eck, Claire Cui and Yonghui Wu for helpful discussions, Krishna Srinivasan, Samira Daruki, Nan Du and Aashi Jain for help with data preparation, Chao Jia, Zhen Li, Jonathan Shen, Colin Raffel and Sharan Narang for assistance on experimental settings, and others in the Google Brain team for support throughout this project.

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

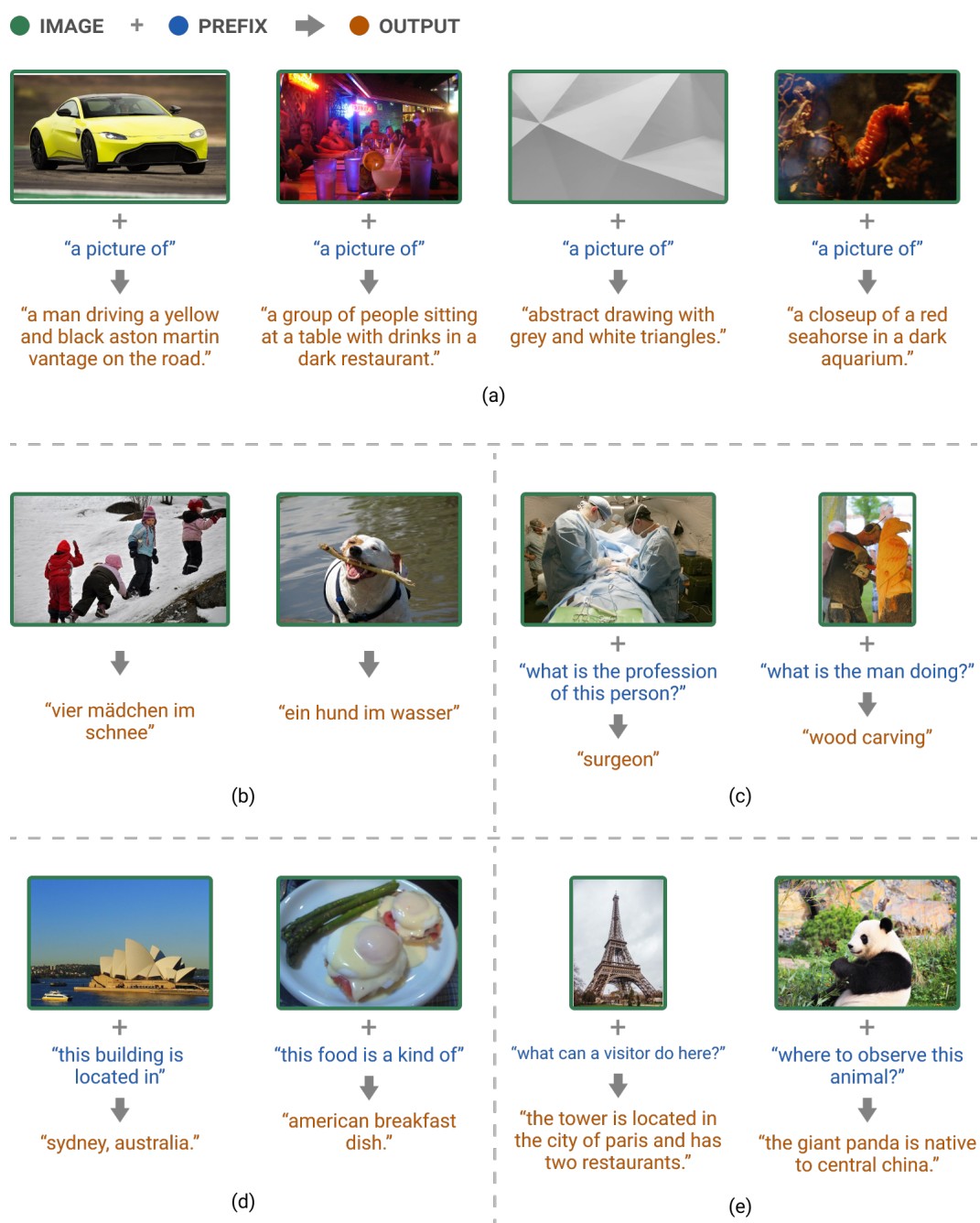

Figure 2: Generated examples of SimVLM of various applications: (a) zero-shot image captioning (b) zero-shot cross-modality transfer on German image captioning (c) generative VQA (d) zero-shot visual text completion (e) zero-shot open-ended VQA.

## A    GENERATED EXAMPLES

Examples generated by SimVLM of various types are shown in Figure 2. We use either image-only or image-text prefix inputs in the encoder, and use the decoder to generate suffix text.

|  | CoLA | SST-2 | RTE | MRPC | QQP | MNLI | QNLI | WNLI |
|---|---|---|---|---|---|---|---|---|
| BERT | **54.6** | **92.5** | 62.5 | **81.9/87.6** | **90.6/87.4** | **84.2** | **91.0** | 48.8 |
| VisualBERT | 38.6 | 89.4 | 56.6 | 71.9/82.1 | 89.4/86.0 | 81.6 | 87.0 | 53.1 |
| UNITER | 37.4 | 89.7 | 55.6 | 69.3/80.3 | 89.2/85.7 | 80.9 | 86.0 | 55.4 |
| VL-BERT | 38.7 | 89.8 | 55.7 | 70.6/81.8 | 89.0/85.4 | 81.2 | 86.3 | 53.1 |
| VilBERT | 36.1 | 90.4 | 53.7 | 69.0/79.4 | 88.6/85.0 | 79.9 | 83.8 | 55.4 |
| LXMERT | 39.0 | 90.2 | 57.2 | 69.8/80.4 | 75.3/75.3 | 80.4 | 84.2 | 46.0 |
| SimVLM$_{base}$ | 46.7 | 90.9 | 63.9 | 75.2/84.4 | 90.4/87.2 | 83.4 | 88.6 | **58.1** |

Table 7: Text-only task performance on the GLUE benchmark (Dev set). Results for BERT and other VLP methods are obtained from Iki & Aizawa (2021). The overall best result is **bolded** while underline signifies the best VLP model.

## B  EXPERIMENTAL DETAILS

### B.1  PRETRAINING

Our models are pretrained according to the methodology described in Section 3. For the Transformer, each variant follows the same setting as its corresponding ViT variant. For the Conv stage, we use the first three blocks (excluding the Conv stem) of ResNet-101 and ResNet-152 (He et al., 2016) for our Base and Large models respectively, and a larger variant of ResNet-152 with more channels for the Huge model (matching its hidden dimension size). We always use a fixed patch size of 16×16. During pretraining, we utilize the resolution of 224×224, resulting in a patch sequence of length 14×14 as visual tokens. For the textual input, we use a vocabulary size of 32,000 and a max sequence length of 256 in both the encoder and the decoder. We also share parameters between the embedding and the decoder softmax output layer (Press & Wolf, 2016). All parameters are shared across visual and textual inputs except the Conv stage and positional embeddings.

We pretrain on large-scale web datasets for both image-text and text-only inputs. For joint vision and language data, we exploit the training set of ALIGN (Jia et al., 2021), which contains about 1.8B noisy image-text pairs. Notice that we do not use any extra data preprocessing or filtering, except simple random resized cropping. For the text-only copora, we use the Colossal Clean Crawled Corpus (C4) dataset presented in Raffel et al. (2019) and followed their preprocessing steps. The dataset contains about 800GB of web crawled documents.

All models are pretrained for about 1M steps from scratch to optimize for the single PrefixLM objective in Eq.3. We use the AdamW optimizer (Loshchilov & Hutter, 2017) with $\beta_1 = 0.9, \beta_2 = 0.999$ and weight decay of 0.01. We warm up the learning rate for the first 2% of updates to a peak value of $5\times10^{-4}$, and then linearly decay it afterwards. Dropout is not used during the pretraining stage. We mix the two pretraining datasets within each batch, which contains 4,096 image-text pairs and 512 text-only documents, sharded across 512 TPU v3 chips (Jouppi et al., 2017).

### B.2  FINETUNING

After pretraining, our model is finetuned on various downstream tasks. Similar to the pretraining stage, we use the AdamW optimizer with the same Beta values, while we tune the learning rate in $\{1\times10^{-5}, 2\times10^{-5}, 5\times10^{-5}\}$. We also enable regularization methods of Dropout (set to 0.1) and stochastic depth (only applied to Conv stage and encoder with a fixed dropout rate of 0.1) (Huang et al., 2016) during the finetuning stage. Following standard practice, we use the corresponding dev split to find the best setting and report the result on the test split. We consider 5 types of downstream tasks listed below:

**Visual question answering:** This task requires the model to answer questions about input images, and has been the most widely used VL benchmark. Following prior work, we use the VQA v2 (Goyal et al., 2017) and formulate the task as a classification problem over 3,129 most frequent answers in the training set. The raw image and the corresponding question are used as inputs to the encoder and the decoder respectively, and a task-specific linear classifier is trained to predict answer

based on activation corresponding to the last question token from the decoder. We use a resolution of 480×480 for the image and all positional parameters are adapted using linear interpolation.

**Visual entailment:** The SNLI-VE (Xie et al., 2019) dataset is adapted from SNLI (Bowman et al., 2015), which is originally designed to predict the relation between a premise sentence and a hypothesis sentence as either entailment, neutral or contradiction, a task known as natural language inference (NLI). For the VL variant, the premise is based on the content of an image rather than textual descriptions. We finetune SimVLM similarly to VQA, such that the image and the sentence are fed to encoder and decoder separately, and the classifier is trained to predict the three relations.

**Visual reasoning:** The NLVR2 (Suhr et al., 2018) dataset tests the model's ability of jointly reasoning over the language and multiple images by asking whether a textual description is true based on a pair of two images. Following Zhang et al. (2021), we create two input pairs, each consisting of one image and the textual description, and generate output embeddings for both using the same setup above. The two embeddings are then concatenated for final prediction.

**Image captioning:** The captioning task requires a model to generate natural language descriptions of input images. We consider two datasets CoCo (Chen et al., 2015) and NoCaps (Agrawal et al., 2019), both finetuned using the CoCo training data. For SimVLM, it is straightforward to first encode the image in the encoder and then generate captions using the decoder. Note that in contrast to prior work that apply task-specific tricks such as CIDEr optimization (Rennie et al., 2017), our model is trained with naive cross-entropy loss only.

**Multimodal translation:** The goal of multimodal translation is to translate image descriptions in source language to target language, for which image inputs can be taken advantage of as grounding signal. We train and evaluate on the Multi30k (Elliott et al., 2016) dataset. We utilize the PrefixLM described in previous sections such that the source sentence, together with the image inputs, are fed to the encoder, which will be translated to the target language by the decoder.

## C  MODEL PERFORMANCE ON LANGUAGE-ONLY TASK

We compare our model with prior VLP methods on natural language understanding (NLU) tasks on the GLUE benchmark (Wang et al., 2018) in Table 7.

## D  ERRATUM

We found an error in reporting the zero-shot COCO evaluations in the first version of this paper. This mistake does NOT affect all other results and the numbers have been updated. Meanwhile, we also added few-shot results in addition to zero-shot results on both MsCOCO and NoCaps in Table 2, to provide a more comprehensive view of capacities in SimVLM models. Hence, our main claims and conclusions still hold.

