# OpenReview forum: "SimVLM: Simple Visual Language Model Pretraining with Weak Supervision"
_ICLR.cc/2022/Conference — ICLR 2022 Poster_

### Official Review · Reviewer_PHG8 · 2021-10-20

**Correctness:** 3
**Technical Novelty And Significance:** 3
**Empirical Novelty And Significance:** 3
**Recommendation:** 6
**Confidence:** 5

**Details Of Ethics Concerns:**

The experiments in this paper heavily rely on the internal dataset (ALIGN, Jia et al., 2021), which is not accessible from outside of the associated group. This practice crucially prevents reproduction and evaluate the presenting work. For that reason, it also limits a fair comparison with the other competing methods. Unfortunately, the authors do not deliver the result using a publicly available dataset for a fair comparison.

**Main Review:**


### Strength

**Simplified pretraining procedure.** The most important point is to simplify a pretraining procedure. In recent works, MLM and image-text matching (ITM) losses are repeatedly used and additionally propose dataset-specific losses to exploit inductive biases. Unlike this trend, this work counters the general belief and gives a new insight to use the PrefixLM for vision-language pretraining (VLP) as a standalone objective. However, this paper's argument seems to be fragile for the reasons in the below weakness points.

**Generalization and transfer knowledge.** The PrefixLM is a generative modeling loss compared to the MLM's classification loss. For this reason, it may be keen on zero-shot generalization. They show competitive or better performance on zero-shot image captioning, zero-shot cross-modality transfer, and open-ended visual question answering tasks.

### Weakness

The PrefixLM is the paper's central argument, aside from the weak supervision dealt with in ALIGN (Jia et al., 2021) and others (e.g., CC 12M, Changpinyo et al., 2021). However, this paper lacks solid validation of their argument, as summarized below.

**W1. Can the PrefixLM replace MLM?** The critical ablation study to see the effectiveness of PrefixLM compared to MLM is 1) limitedly shown on VQA and image captioning in Table 6, a limited number of downstream tasks compared to the other experiments, although it is a critical study to validate their argument. Moreover, 2) this ablation study is performed for text-only data, not vision-language data in Section 4.4 Ablation Study. For this reason, their argument is fragile to be nullified in the further validation in the current status.

- Could you provide any evidence that the PrefixLM is better than MLM in the controlled experiment (ablation study) for multimodal data?

**W2. Can the PrefixLM be used as a standalone objective?** For instance, the ITM loss is known to be effective combining with MLM (e.g., ViLT, Kim et al., 2021). They argue that a single objective is enough, but there is no validation of whether this is true. The natural questions on this matter include combining with ITM or other dataset-dependent losses would be helpful or result in mediocre performance.

- Could you provide any evidence that PrefixLM + ITM method is not significantly different from PrefixLM only method? If not (PrefixLM + ITM is better), what would be the advantage of using the PrefixLM only?

**W3. Fair comparison issue with the ALIGN dataset.** The crucial problems in the experiments are that 1) it is irreproducible for its inaccessible collected datasets outside of the associated groups, and 2) it is hard to assess the contribution in the state-of-the-art comparison due to different pretraining datasets, different model architectures, and pretraining objectives. The controlled experiment still has the problems as described in W1.

- For the reproducibility, could you provide the scores pretraining on WIT (Srinivasan et al., 2021) only?


### Writing

- In the RHS of Eqn.3, if $t=T_p$, is $P(x_{T_p}|x_{[T_p,T_p]}, x_{< T_p})$ the intended one? Should $x_{[T_p,T_p]}$ be null?
- It is ambiguous that "which decouples encoding from generation is conducive to the improvement of downstream task" in the 1st paragraph, Section 3.3. What do you mean by "decouple"?
- In the 2nd paragraph of Section 3.3, "we additionally add 2D relative attention for the image patches within transformer layers," which needs some elaboration for reproducibility and self-contained explanation.
- Using C4 dataset was crucial? Why do you choose to use C4 additionally?
- In the 4.1 Setup, you mentioned that "512 text-only documents are in each batch," but, in this reading moment, cannot follow the context.
- In Table 2, "'Pre.' indicates the model is pretrained" on what? Are you mentioning that the corresponding models are pretrained on CoCo or NoCaps? Here, all SimVLM is also pretrained on both ALIGN and C4 datasets, but you don't say all your model is pretrained.
- In 4.3.1, please be aware that your Figure 2 is placed in Appendix and indicate explicitly.
- In Table 2, could you specify the meaning of each row section? Why are two SimVLM_{huge} scores different in the 2nd and 3rd sections?
- Explanation on transformer decoder is simply skipped. Could you provide implementational details for reproduction?

**Summary Of The Paper:**

This paper proposes a Prefix Language Modeling (PrefixLM) objective for a pretraining procedure for multiple vision-language downstream tasks and zero-shot evaluations. They argue that it successfully replaces the masked language model (MLM). This work follows the notion of weak supervision suggested by ALIGN (Jia et al., 2021), having noisy image captions and holistic labels. Surprisingly, this simple pretraining approach achieves new state-of-the-art on "a wide range of discriminative and generative vision-language benchmarks," and shows "strong generalization and transfer ability" in zero-shot settings.

**Summary Of The Review:**

This paper shows that a simple pretraining method, the PrefixLM, using two datasets from the dataset of ALIGN (Jia et al., 2021) and the Colossal Clean Crawled Corpus (C4) dataset (Raffel et al., 2019) (optionally, along with WIT (Srinivasan et al., 2021)) achieves new state-of-the-art in various vision-language benchmarks. However, this work is narrowed to achieve the state-of-the-art performances while failing to show 1) better performance of the PrefixLM than MLM, 2) relationships with the other multimodal pretraining losses (e.g., ITM) 3) in the controlled experiment in terms of pretraining dataset and model architecture. The weaknesses W1-3 hinder this paper's recommendation since the main argument of the PrefixLM is in a vague position.

---

> ### Author Response · Authors · 2021-11-22
> **Responses to Reviewer PHG8**
>
> Thank you for your comprehensive review and valuable feedback! To clarify, our manuscript does **NOT** aim to claim that 'PrefixLM is better than MLM' and pursuing SOTA results is not our main goal. Instead, as stated in the introduction, our goal is to explore a simpler generative VLP framework that has **good zero-shot potential** while remaining **competitive on standard VL benchmarks**. Our results verify that the generative pretraining paradigm can be a promising direction and we hope our work may inspire future research. We have updated our manuscript to better reflect our claim and motivation.
>
> Beside, we address your comments one by one as following:
>
> [PrefixLM vs MLM]
> As stated above, we never intend to argue that PrefixLM is better than MLM. Rather, inspired by GPT3, we are curious to explore the limit of similar pretraining paradigms and probe its zero-shot generalization potential on VL tasks. Our hope is that PrefixLM can match the performance of MLM and enable zero-shot behaviors.
>
> In fact, in our preliminary experiments we found them to be comparable for finetuned results. This is consistent with findings in NLP where these two objectives perform competitively on natural language understanding tasks (e.g. BERT vs T5/GPT). Analogous to the comparison between BERT and T5/GPT (such that each has its own focus and we do not say one is better than the other), we are not arguing that one VLP objective should replace the other but just to demonstrate the effectiveness of prefix modeling on VL tasks.
>
>
> [PrefixLM as a standalone objective]
> Actually we have experimented with PrefixLM+ITM and found no significant difference (PrefixLM+ITM gets ~67 on VQA compared to 67.43 for PrefixLM-only in table 6). However, we think possibly some auxiliary losses can further improve the model quality with more careful and proper tuning, which we consider as an important future direction (updated in our conclusion).
>
> We have also tried a few other objectives such as image feature regression but they didn't provide extra gains either. This is potentially due to underfitting of the model or specific implementation/tuning problem.
>
> [Data scaling]
> We have pretrained SimVLM_small with the smaller cc3m dataset (see table 6) and the performance gap is not big. On test-dev of VQA, this variant obtains 70.45 which is comparable to MLM based approaches such as ViLT (given the smaller model sizes). This further verifies that PrefixLM is capable of matching the performance of MLM on standard VL benchmarks.
>
> That said, recent work such as GPT3/CLIP suggest that zero-shot learning emerges with data scaling and thus we chose the large-scale weakly aligned data. We believe this is an important ingredient towards zero-shot generalization and we will further study this in a future work.
>
> [Data accessibility]
> We agree that data accessibility is an open issue in the research community. Many impactful work [1-7] also suffer from this constraint since data releasing can be controversial as well (e.g. [8,9]). Unfortunately, fixing this problem is beyond our control, and we believe the whole community might have to work together to address this issue.
>
> [Eq. 3]
> Yes, this is intended. When t = T_p, no text should be used in the encoder and thus it is null.
>
> ['decouple']
> Here we mean compared to the decoder-only model which conducts encoding and generation in a single transformer, the encoder-decoder model has encoder for encoding and decoder for generation.
>
> [C4 dataset]
> ALIGN is a weakly aligned dataset that is extremely noisy especially in the text modality. We add C4 to improve the language understanding and generation to enable zero-shot decoding.
>
> We also appreciate your comments on writing. We have addressed these issues and we will update additional training details in our final version.
>
> [1] Brown et al.. Language Models are Few-Shot Learners.
> [2] Radford et al. Learning Transferable Visual Models From Natural Language Supervision.
> [3] Ramesh et al.. Zero-Shot Text-to-Image Generation.
> [4] Sun et al.. Revisiting Unreasonable Effectiveness of Data in Deep Learning Era.
> [5] Xie et al.. Self-training with Noisy Student improves ImageNet classification.
> [6] Dosovitskiy et al.. An Image is Worth 16x16 Words: Transformers for Image Recognition at Scale.
> [7] Mahajan et al.. Exploring the Limits of Weakly Supervised Pretraining.
> [8] Schuhmann et al.. LAION-400M: Open Dataset of CLIP-Filtered 400 Million Image-Text Pairs.
> [9] Birhane et al.. Multimodal datasets: misogyny, pornography, and malignant stereotypes.

---

### Official Review · Reviewer_Z7A6 · 2021-11-02

**Correctness:** 3
**Technical Novelty And Significance:** 3
**Empirical Novelty And Significance:** 4
**Recommendation:** 8
**Confidence:** 4

**Main Review:**

#### [Strengths]
- Overall, I like this paper. The proposed method is very simple but effective and practical because the pretraining task is not complex and the used weakly aligend image-text data are relatively easy to acquire compared to those used in other methods such as CLIP.
- Despite its simplicity, the performance are promising for various fine-tuning tasks.
- The paper is easy to understand.
- The experiments are thorough and support their hypothesis.

#### [Weakness]
Despite many strenghs of this paper, there are some room to be improved. My score is between 6 and 8 due to the issues below:
- In introduction, the authors argued "These design choices complicate the pretraining protocol of VLP, creating a bottleneck for further quality improvement." and "these methods mainly focus on specific tasks of consideration and thus may not serve as a generic pretraining-finetuning representation for VL benchmarks.". Are there any explicit evidences or refences for this arguement?
- Figure 1 can be improved. When zooming up, some texts are not clear.
- Reproduciblity is limited. The limited reproducibility by other research groups results from data issue. It is not trivial to construct 1.8B paired data even if they are weakly-aligned and train them. Also, I guess that the promising results of a simple pretraining task and architectures might benefit from large-scale size of its training data. Therefore, the ablation experiments on various training data sizes (e.g. 10%, 30%, 50%, and 100%) will be helpful for understanding the effects of the data size on the performances. If smaller size data can provide comparable results, the contributions of this work will be much enhanced.
- Presenting parameter sizes of each model variant will help to understand the results.
- Table 2 is not easy to read. In Table2, what is the difference betwwen the second and the third row groups? I guess the second is for zero-shot and the third one is for fine-tuning. This should be explicitly clarifed in Table for readability. Also, why is SimVLM_large checked only as pretraining?
- The results in the main paper and the appendix are explained in the main manuscript. For example, even if Figure 2 and Table 7 exist in the appendix, there is no information on those. This harms the readability.
- Table 3 can also be improved. Fine-tuned datasets and metrics are located at the same row. Image Masking results also might harm the readability.
- Why the perfomances of SNLI-finetuned is better than those of SNLI-VE-finetuned? The authors need to clarify this reason.
- Multimodal learning can provide more robust performance on different domain data like zero-shot image classification on line-draw images of CLIP. SimVLM also present more robust image classification performances on different domain data? In addition to classification accuracy, these results can enhance the contribution of this paper, if possible.
- In Ablation study, why used SimVLM_small? How about the possibility that poor performance of decoder-only model resulted from its smaller model size?
- Minor issues (typos)
  - In 3.4 in p4, "with raw image patch inputs." . --> , (comma)
  - in 4.1 in p, "Huge".: . --> , (comma)
  - Karpapth --> Karphathy


**Summary Of The Paper:**

This paper proposes a simple but effective image-text multimodal representation learning method that leverages a transformer-based encoder-decoder using a simple prefix langemodel as pretraining task from large-scale noisy image-text aligned data. As fine tuning tasks, the authors evaluate their method (SimVLM) on VQA, NLVR2, SNLI-VE, CoCo caption, NoCaps, and Multi30k. With extensive experiments, this work presents promising few-shot and zero-shot performance results outperforming previous models.

**Summary Of The Review:**

Overall, this is a good paper but there are some issues that should be improved, in particular, reproducibility and table for readability.

---

> ### Author Response · Authors · 2021-11-22
> **Responses to Reviewer Z7A6**
>
> Thank you for your comprehensive review and valuable feedback! We address your comments one by one as following:
>
> [In introduction, the authors argued...]
> (1) As shown by [1], the performance of existing models largely depends on the object proposal module and thus this could potentially be a bottleneck for further improvement (e.g. scaling with cheap data). (2) Recent models trained with weakly labeled data show strong zero-shot performance but may not perform well when finetuned on standard VL benchmarks. For example, CLIP works well on zero-shot image classification but not quite so when directly finetuned on VQA [2] while DALL-E only focuses on image generation.
>
> [Pretraining with smaller datasets]
> We agree that data accessibility is an open issue in the research community. To better understand the effect of data sizes, we compare models trained with 10% of ALIGN and cc3m in Table 6 . While the scaling of data does improve the model quality, the gap is quite small for finetuned performance. However, we do believe large-scale weakly aligned data has better zero-shot potential, as suggested by [3,4].
>
> [Table 2 clarification]
> Thank you for pointing this out! Here we checked 'Pre' or 'Sup' for the whole block instead of just a single row. So we actually meant to indicate that the second block is for zero-shot inference (only pretraining but no further finetuning on the downstream task). We have updated the caption with further explanation.
>
> [SNLI-finetuned vs SNLI-VE-finetuned]
> This is primarily due to dataset preprocessing. SNLI-VE and SNLI have different ways of splitting the dataset, and thus there are some overlaps between the training split of SNLI and the testing split of SNLI-VE. Even though the exact testing examples are not included in the training set since we are transferring from text-only data to image-text data, we suspect that this may still benefit the model and therefore we additionally include the results trained on SNLI-VE-text.
>
> [Robust cross-domain performance]
> We are currently working on image classification evaluation as our next steps. However, SimVLM does show better generalization across domains. For instance, it can transfer to the unseen domain of CocoCaption (Table 2) and unseen answer types in VQA (Table 4) in a zero-shot manner. It can also generalize from MNLI to SNLI-VE and sometimes even across the language boundary (Table 3).
>
> [SimVLM_small vs decoder-only model]
> We use smaller sized models primarily for efficiency concerns as pretraining variants of larger models can be costly. For the decoder-only models, we actually have evaluated them at a larger scale (up to the size of BERT_Large) and results are consistent. To rule out the factor of model size discrepancy, we always match their sizes, including those in Table 6.
>
> We have also addressed other readability issues as best as we can in the updated manuscript given the space limit. Thank you for your kind suggestions!
>
> [1] Zhang et al.. Vinvl: Revisiting Visual Representations in Vision-Language Models
> [2] Shen et al.. How Much Can CLIP Benefit Vision-and-Language Tasks?
> [3] Brown et al.. Language Models are Few-Shot Learners.
> [4] Jia et al.. Scaling Up Visual and Vision-Language Representation Learning With Noisy Text Supervision.

---

> > ### Comment · Reviewer_Z7A6 · 2021-11-27
> > **Thank you for response**
> >
> > Dear author,
> >
> > Thank you for your efforts. Most of my concerns were allivated. I recommend that all unclear points I raised would be clarified by adding references or more description in the final version. Thank you for sharing good research work again.
> >
> > Because my score was betwen 6 and 8 as I mentioned, I will keep my score as 8.

---

### Official Review · Reviewer_wzgn · 2021-11-08

**Correctness:** 3
**Technical Novelty And Significance:** 2
**Empirical Novelty And Significance:** 3
**Recommendation:** 6
**Confidence:** 4

**Main Review:**


Pros:

1. The empirical results are strong. SimVLM takes an important step towards less human-label-dependent V&L models.

2. Interesting results on some zero-shot tasks. In Sec 4.3.2, the authors introduce a new setting: train on a similar single-modality task and transfer directly to a multi-modality task. This is a new yet promising way of transferring to new V&L tasks.


Cons:


1. Dataset deduplication. I do not see if the authors check whether downstream tasks data are present in the pre-training dataset. It is recommended to check the overlap.

2. Image caption / SNLI-VE and VQA experiments are all grouped into the zero-shot setting. They are technically "zero-shot" but they actually differ quite a lot from others. SNLI-VE & VQA experiments are also not the "full zero-shot" setting where we simple take a model and directly get the desired output. I would recommend being more careful about the wording.

3. For the experiment on SNLI-VE in Table 3, I think the same method could also be done for UNITER (i.e. train on SNLI and then transfer to SNLI-VE). It is an informative baseline to have.


Minor points:

4. Could the cross-modality experiments be extended to other tasks such as VQA? Could we train on  some similar text-only tasks and then transfer to VQA? The SNLI-VE is a good sign but SNLI-VE is a three-way classification task and is perhaps easier to transfer to than VQA.

5. Efficiency concern. Training on CLIP/ALIGN-level data is costly. A seemingly more efficient way to use trained CLIP/Align model as bottom backbones and train a V&L transformer on top (e.g., Shen et al., 2021). With the same model scale (base), the VQA score is not so far apart (78.14 v.s. 76.48).
Shen et al., How Much Can CLIP Benefit Vision-and-Language Tasks?

6. True Zero-Shot VQA? I wonder what prevents the authors from directly testing on VQA. One could give the image + question as the prefix and then calculate the generation probability for every answer candidate from the 3,129 answer pool. This would serve as a valuable reference point.


**Summary Of The Paper:**

The paper proposes to pre-train a generative language model conditioned on a visual input on billion-scale web image-text data. Such a model can be then transferred to various vision-and-language tasks with ease. SimVLM establishes new SotA on several new tasks and shows promising zero-shot capacity in certain tasks.



**Summary Of The Review:**

Overall, I think the cons are points that could be addressed and do not hurt the main merit of the paper. Thus I recommend acceptance but I would recommend addressing the cons if possible.

---

> ### Author Response · Authors · 2021-11-22
> **Responses to Reviewer wzgn**
>
> Thank you for your comprehensive review and valuable feedback! We address your comments one by one as following:
>
> [Dataset deduplication]
> We have checked that. We are using two existing datasets ALIGN [1] and C4 [2], and they have already checked and removed duplicates or near-duplicates of popular downstream tasks. For example, the vision-language tasks considered in our paper share the same data sources as those in [1] and thus this should not be an issue.
>
> [Zero-shot wording]
> Thank you for pointing this out! We agree the term 'zero-shot' has been a bit ambiguous in our paper. In particular, it could refer to zero-shot inference, zero-shot cross-modality transfer and zero-shot generalization on unseen examples. Although these settings are different, they all share the goal of generalizing to out-of-domain  knowledge (be it a downstream task, a different modality or a test example different from the training set) without further finetuning and thus we refer to them as 'zero-shot generalization'.  We have updated our paper to articulate their differences.
>
> [UNITER zero-shot transfer]
> We agree that this new setting is applicable to some existing models. Specifically, we have updated results for UNITER in Table 3. We hypothesize three factors that make SimVLM better on this zero-shot transfer setting: (1) SimVLM does not utilize any modality embedding as UNITER and thus blurs the modality boundary. (2) The enc-dec architecture design is more flexible and better suited for zero-shot transfer. (3) We adopt large-scale, domain-agnostic pre-training datasets, which potentially improve cross-modality generalization.
>
> [Zero-shot transfer on VQA]
> We actually have considered transfering to VQA but couldn't find a language-only QA dataset that is similar in nature. Most existing NLP QA datasets are of the form of span prediction or multiple choice, while other open-ended QA datasets are typically conditioned on knowledge base instead of evidence. Therefore, we considered the generation task of zero-shot cross-lingual cross-modality transfer, which is more demanding than NLI tasks.
>
> [Training from scratch vs warm start with pretrained models]
> We agree that adapting pretrained models is another promising approach for vision-language tasks. However, pretraining from scratch relieves the constraint of inductive bias of existing models and thus allows us to explore new directions. We have added a discussion in the related work.
>
> [Fully zero-shot VQA]
> We didn't consider this setting mainly for two reasons: (1) Using the predefined 3129 answer set violates the zero-shot setting since they are obtained through the training split, and thus this should not be considered as true zero-shot open-ended VQA. (2) Besides, decoding generation probabilities with 3129 candidates (alone with image+text prefix tokens) on the test set of ~400k examples is computationally heavy and prohibitively slow.
>
> [1] Jia et al.. Scaling Up Visual and Vision-Language Representation Learning With Noisy Text Supervision.
> [2] Raffel et al.. Exploring the Limits of Transfer Learning with a Unified Text-to-Text Transformer.

---

### Public Comment · ~Xinsong_Zhang1 · 2021-11-15
**How much gain the data brings? Thank you.**

Dear authors,
Thanks for your great work!  The paper is easy to understand with many insights.
I have similar questions with Reviewer Z7A6 on the benefit of the large corpus (1.8B). I want to know how much gain the data brings, and would appreciate it if you could provide some ablation studies on various training data sizes. Taking a comparable data size as other commonly used pre-training corpora (such as mscoco+visual genome+cc3m, around 4M image-text pairs) into consideration is also helpful.

By the way, will you release the code and pre-trained models in the future?

Thanks!

---

### Decision · Program_Chairs · 2022-01-20

**Decision:**

Accept (Poster)

**Comment:**

This paper presents SimVLM, a simpler generative VLP framework with billion-scale web image-text data, which has good zero-shot potential while remaining competitive on standard VL benchmarks. SimVLM achieves SotA on several tasks and shows promising zero-shot capacity in certain tasks. Most of the reviewers liked the work; they had concerns about data scaling, but the authors showed that SimVLM_small with the smaller cc3m dataset does not drop performance too much (although the large data scaling with their large-scale weakly aligned data is still important to achieve good zero-shot learning). All reviewers also mentioned the strong concern of reproducibility and data accessibility, so we encourage the authors to address this as clearly as possible via releasing models and safely-cleaned/anonymized data subsets, etc.